# Construct Validity of the Attitudes towards Maghrebis in Education Scale (AMES)

**DOI:** 10.3390/ijerph19127303

**Published:** 2022-06-14

**Authors:** Miguel Ángel Albalá Genol, Edgardo Etchezahar, Juan Ignacio Guido, Joaquín Ungaretti

**Affiliations:** 1Department of Evolutionary Psychology and Education, Faculty of Teacher Training and Education, Universidad Autónoma de Madrid, 28049 Madrid, Spain; edgardoetchezahar@psi.uba.ar; 2Faculty of Education, International University of Valencia, 46002 Valencia, Spain; joaquinjesus.ungaretti@campusviu.es; 3Faculty of Psychology, University of Buenos Aires, Buenos Aires C1053ABH, Argentina; juanguido@psi.uba.ar; 4National Scientific and Technical Research Council, Buenos Aires 1428, Argentina

**Keywords:** interculturality, education, empathy, migrants, scale

## Abstract

The current article describes the validation of the Attitudes towards Maghrebis in Education (AMES) scale in the Spanish context and examines the relation with other psychosocial variables. A Spanish sample of 807 participants aged between 18 and 80 years old completed the AMES. The analyses were performed using CFA, mainly considering the statistical indices of CFI, RMSEA and Pearson’s correlation. The external validity of the scale was tested using measures, such as social dominance orientation (SDO), empathic concern, warmth, competence and contact with Maghrebis. The results indicated that the AMES showed an adequate fit to the data. The AMES was found to be negatively correlated with SDO, whereas it was correlated positively with the rest of the variables. The results demonstrate that the AMES can be used as a reliable measure to evaluate attitudes towards Maghrebi migrants in education in Spain. The implications of the psychosocial variables studied are discussed as possible factors to consider for promoting more intercultural socio-educational environments.

## 1. Introduction

In the last two decades, a study on the social well-being and the socio-educational inclusion of migrant students took more relevance due to significant changes (e.g., economic, political and educational, among others) that transformed part of the previous values, beliefs and forms of coexistence [1]. Against this backdrop, inclusion mechanisms became more volatile and diffuse, with migrant people engaged in new modes of transnational coexistence. The different crises experienced in recent decades (e.g., the international economic recession in 2008, the global socio-sanitary crisis due to COVID-19, and numerous emerging refugee and migrant crises around the world) have increased the relevance of inequality over cultural diversity, emphasizing the differences in status between migrant groups in Southern Europe [2]. In the same way, in a Latin American context, there is no common operational structure linked to the exercise and guarantee of social migrant rights [3]. Thus, the social dimension present in education is strongly related to issues, such as social justice and citizenship, being a relevant field for the study of diversity, multicultural perspectives or the situations of migrant students [4]. Many of these changes have led to numerous economic, sociopolitical and humanitarian crises that have increased the migratory flows to developed countries, such as Spain. As a result, these countries have become multicultural societies, however, they have not always generated frameworks for intercultural coexistence [5]. According to the data collected by the Spanish National Institute of Statistics [6] about migratory movements towards Spain, in 2021, the migrant population increased by 2% compared to 2020. The same report indicates that the migrated populations from the Maghreb-region countries (Morocco, Algeria, Libya, Mauritania and Tunisia) are the most numerous in the Spanish territory. Thus, due to the great increase in xenophobic speeches towards Maghrebis in recent years [7], mainly towards people of Moroccan origin, it is important to delve into the psychosocial factors involved in the intercultural attitudes towards the Maghrebi community in Spanish schools and universities.

The evidence of this is that social and educational inclusion comprises a strong psychosocial dimension: negative attitudes towards the presence of migrants in educational communities (teachers, students and families). In this sense, according to the Sustainable Developmental Goals (SDG) of the 2030 Agenda, SDG-4 refers directly to achieving a quality education [8]: “ensure inclusive and equitable quality education and promote lifelong learning opportunities for all”. This goal makes explicit the need for transformation and change in future societies, through a more inclusive education that is oriented towards interculturality and any form of diversity. From this perspective, the educational system must promote upward mobility for the most disadvantaged students, to become a mechanism to eradicate situations of social and educational exclusion. In addition, SDG-3 (Ensure healthy lives and promote well-being for all people) raises the need to guarantee universal access to information and education services [9]. Therefore, access to these educational services also has an impact on the well-being of migrants and, therefore, should be guaranteed.

Different studies have shown that, in Spain, migrants are the most affected by prejudice and discrimination [10,11]. However, considering the ethnic category as a source of discrimination and exclusion, few studies have focused on objectives beyond descriptive ones [12,13]. In addition, many of these studies have an approach which is based on the study of the general migrant community, showing the difficulties of delving into more specific analyses, such as the possible factors involved in attitudes towards Maghrebis in the educational field. According to [14], this could be due to the lack of measures for evaluating attitudes towards the interculturality and sociocultural diversity of specific migrant groups. These measures would allow for a more detailed understanding of the possible psychosocial variables linked to the development of intercultural attitudes.

### 1.1. Intercultural Attitudes and Their Evaluation

The psychosocial field has identified attitudes as the evaluations that people make (favorable or unfavorable) about certain objects, people or groups. Traditionally, attitudes were considered from the explicit point of view, that is, a reference is made to the self-report that a person makes, or to the direct and clean manifestation that a person makes about a certain object, person or group. The traditional method to evaluate these attitudes has been that of questionnaires and graduated scales. However, other measures were also designed that allow access to people’s internal states and attitudes without being directly consulted about them [15]. These implicit measures would allow a quicker and less conscious evaluation (e.g., “Implicit Association Test” [16]). These two types of attitudes are important, as they have been suggested to guide judgments and behavior. In the present study, it is of interest to analyze and research the factors that allow the promotion of favorable behaviors towards interculturality in education. Thus, when people have enough motivation, time, or cognitive resources at their disposal, explicit attitudes should guide their behavior [17].

The notion of interculturality, according to [18]: “involves the relationships established between people and social groups belonging to different cultures and that advocate dialogue and encounter between them, within the framework of a citizenship concept that involves equality of rights” (p. 15). From this perspective, the existing inequality among the migrant group should be an incentive so that, from the socio-educational level, not only is having access to education promoted, but it also helps to build an intercultural coexistence. At this point, the participation of all social groups, inside and outside the educational field, is crucial for improving the inclusive and intercultural aspects. This process of recognition and participation in an increasingly multicultural citizenry means that one of the great challenges in terms of educational inclusion is related to the focus on the migrant population diversity, inside and outside the classroom [19].

In order to assess these topics, the Intercultural Sensitivity Scale-ISS was developed by Chen and Starosta [20], to measure people’s attitudes towards intercultural communication. In its abbreviated version, it was made to include five dimensions: interaction engagement, respect for cultural differences, interaction confidence, interaction enjoyment and interaction attentiveness. These dimensions assess people’s levels of intercultural sensitivity. Both the original and subsequent studies [21] reported adequate reliability indices as well as good internal and external validity. In addition, several relationships were found with other psychosocial variables, such as sexism and psychological well-being. 

Another contribution to the assessment of intercultural sensitivity in the educational context is the questionnaire developed by [22]. This test, given the situation of increasing cultural diversity and the presence of migrants, explores three dimensions: school role, teacher’s role, and a forward-looking perspective. However, no evidence of reliability or validity of the test was reported. 

From a broader perspective (not specifically aimed at the educational field) and focused on building citizenship, Palou and Marín [23] developed a questionnaire to analyze intercultural coexistence and participation. After passing through an interjudge agreement development period, adequate internal consistency indices were reported. More recently, the interculturality scale was developed [24], designed and applied in the context of future educational professionals’ training. The authors designed an extensive scale composed of 56 items, with a Likert response format, structured in five dimensions: immigration phenomenon in the educational context, cultural diversity and its link with school coexistence, interculturality as a pedagogical proposal, resources and pedagogical support in terms of attention to diversity and practical development of interculturality. This scale was validated through a review by expert judges with extensive teaching experience, as well as through its reliability and factorial structure validity. 

Nevertheless, the mentioned developments, although they delve into highly relevant aspects related to intercultural attitudes towards migrants, do not take into account the particularities that occur with specific populations. This aspect is important, since different studies [25,26] have shown that intercultural attitudes and sensitivity towards cultural diversity are not expressed uniformly towards all migrant groups. Therefore, it is possible to develop favorable intercultural attitudes toward certain groups with some forms of cultural diversity, and negative attitudes toward others. For these reasons, the use of tests that refer to a specific group as the object of evaluation is essential.

### 1.2. Interculturality as an Educational Paradigm to Face Cultural Diversity: Psychosocial Variables Involved

All socio-educational processes are related to the external factors linked to social coexistence and beliefs, attitudes and behaviors that, in many cases, violate the basic principles of interculturality in educational communities. This is why various educational and psychosocial variables may be related to attitudes towards migrant diversity in education and towards a more intercultural approach. Based on the main purpose of analyzing the psychometric characteristics of the interculturality scale, it is essential to investigate the other variables that have been shown to be linked to it in order to test its external validity. According to [27], the development of attitudes is not individual but a combination of personal circumstances, values and the environment (political discourses, migration and social policies, the media, civil society and education, among others). Regarding attitudes towards migrants, different studies [28,29] have shown a great influence on psychosocial, political-ideological and economic factors. Among them, social dominance, empathy, contact and stereotypes stand out, allowing the development of more or less favorable attitudes towards the interculturality of migrants as a whole, including Maghrebis [30].

Regarding social dominance orientation (SDO) [31], this focuses on analyzing social functioning and the existing hierarchies that are structured based on different variables, such as origin, race, religious beliefs, gender or social class. In this sense, it has been shown that SDO is related to multiple forms of prejudice [32], particularly prejudice towards migrants [33] and racism [34]. Likewise, another central variable for the study of interculturality is empathy, which is defined as the ability of people to put themselves in another’s place [35]. More empathetic people used to have a greater sensitivity to understanding the vulnerable situation of the migrant population, therefore, it is directly related to attitudes towards interculturality. Another central variable for the analysis of interculturality is the level of contact a person maintains with the migrant population since, as the evidence shows, higher levels of contact decrease the prejudice levels towards migratory groups. Finally, in line with the above, the levels of warmth felt and competence perceived in a migrant group have a strong relationship with the stereotypes towards them [36]. The higher the levels of warmth and competence towards a migratory group, the more difficult it will be to develop positive attitudes towards interculturality. 

Based on the above, the aim of this study is to develop a scale to assess attitudes towards Maghrebi teachers and students in the educational field. Moreover, to analyze the relationships between attitudes towards interculturality and social dominance, empathy, contact with the migrant group and stereotypes of warmth and competence.

## 2. Method

### 2.1. Participants

The present study involved a sample of 807 participant residents in Madrid (Spain). The participants were between 18 and 80 years old. The mean age of the entire sample was 53.09 (*SD* = 13.85); 53.5% were female (n = 432) and 46.5% were male (n = 365). According to the educational level, 6.8% of the participants had completed primary education, 16% had completed secondary education, 20.7% completed short-cycle tertiary education, 38.8% completed undergraduate studies and 17.7% completed postgraduate studies. 

### 2.2. Instruments

An ad hoc questionnaire was created and included the following variables: 

*Attitudes towards Maghrebis in Education Scale (AMES).* The scale was composed of seven items, which linked Maghrebi migrants with different educational scenarios. The items involved different educational agents (both migrant students and teachers of Maghreb origin), including cultural aspects and also the role of the State in making educational institutions have a more intercultural character (see Table 1). The response format of this scale was a 5-point Likert scale of agreement (1 = strongly disagree, 5 = strongly agree). The higher the score in each item and the total scale, the more favorable the attitudes towards Maghrebi interculturality in the educational field. 

*Social Dominance Orientation (SDO).* A version of the original scale [37], adapted and validated to the Spanish context with a 5-point Likert-type scale, ranging from “strongly disagree” to “strongly agree”, was used [38]. The scale’s ten items were grouped into two dimensions: group dominance and opposition to equality, which together conform to the social dominance orientation construct. The higher scores address the higher social dominance orientation levels. The internal consistency (α = 0.89) of the scale was adequate. 

*Empathic Concern (EC).* To evaluate this variable, the empathic concern (EC) dimension of the Interpersonal Reactivity Inventory (IRI; [39]), validated in the Spanish context, composed of eight items (α = 0.78), was used. The response format of this scale was a 5-point Likert scale of agreement (1 = strongly disagree, 5 = strongly agree).

*Warmth and Competence. The Stereotype Content Model (SCM; [40])* was used to assess the stereotypes attributed to Maghrebis. To measure the attributed warmth, the survey inquired to what extent participants evaluated Maghrebis as “warm”, “friendly” and “good-natured”. In the case of the component scale, it was asked to what extent they were evaluated as “competent”, “capable” and “skillful”. All items were answered on a 5-point Likert scale (1 = not at all, 5 = extremely).

*Contact with Maghrebis.* Participants were asked about their contact with Maghrebis and their frequency of contact. A 5-point Likert-type scale was used, ranging from 1 = No contact, 2 = Occasionally, but I don’t usually talk to them, 3 = See them often and interact frequently, 4 = Have friends from that group, 5 = Have relatives from that group. 

*Demographic data.* Information about age, gender, education and the socioeconomic level was also recollected.

### 2.3. Procedure

The participants were invited to participate in the study voluntarily, requesting their informed consent. The data were collected through an online survey through social networks (Facebook, Instagram and Twitter), during January and February 2022. The inclusion criteria to be part of the study were to be of legal age and currently reside in the city of Madrid. Furthermore, participants were informed that the data derived from this research would be used only for academic and scientific purposes under the Organic Law 3/2018, which protects personal data. Additionally, the international methodological standards, recommended by the International Test Commission (ITC) when validating an instrument [41], were followed.

### 2.4. Data Analysis

Statistical analyses were conducted using SPSS 24.0 [42] and EQS 6.1 [43]. First, descriptive statistics for every item were calculated, followed by the analysis of the scale’s reliability and validity. Finally, the relations with sex, age and educational level were studied by running *t*-test analyses, Pearson’s correlation and ANOVA of the variables under study.

## 3. Results 

### 3.1. AMES Item Analysis and Internal Consistency of the Scale

First, the ten items of the AMES were analyzed. The final item wordings, means (*M*), standard deviations (*SD*), skewness (*S*), kurtosis (*K*), item-total correlations (*raj*) and Cronbach’s alpha if an item was deleted (*α.-x*), are displayed in Table 1 for every item.

In general, every item contributed to the overall scale with a relatively high correlation (0.434 < *rjx* < 0.797), and reliability did not improve if an item was removed [44]. The internal consistency of the ATG scale’s adaptation was examined by means of Cronbach’s alpha, which resulted in adequate values (α = 0.88) [45].

### 3.2. Validity Analysis

An exploratory factor analysis (EFA) and a confirmatory factor analysis (CFA) were performed using the seven items of the AMES. The adequacy of the sample was evaluated using the Kaiser–Meyer–Olkin test (*KMO* = 0.888) and Bartlett’s sphericity test (*p* < 0.001). Here, using the mean component analyses, an EFA was calculated with varimax rotation. The obtained model consisted of a single dimension explaining 59.68% of the variance. Afterwards, a CFA was conducted using the maximum likelihood (ML) estimation with Satorra-Bentler’s robust correction (S-B) [46]. Information regarding the model fit is displayed in Table 2.

The results indicate that the proposed model seems to adequately fit the data [45], suggesting the instrument shows acceptable internal validity. 

As suggested by the literature, relations between AMES and other constructs were examined. Hence, Pearson’s correlation coefficients were calculated for AMES, SDO, empathic concern, contact, warmth and competence (Table 3).

As expected, the results indicate a negative and significant association between the AMES and SDO (*r* = −0.655; *p* < 0.01), and are positive with EC (*r* = 0.440; *p* < 0.01), contact (*r* = 0.361; *p* < 0.01), warmth (*r* = 0.744; *p* < 0.01) and competence (*r* = 0.657; *p* < 0.01).

Differences were observed according to the gender of the participants (*t*_(477)_ = 3.932; *p* < 0.001; *Cohen’s d* = 0.36), where women (*M* = 3.50; *SD* = 1.05) scored higher in the AMES compared to the men (*M* = 3.11; *SD* = 1.11). On the other hand, no significant differences were observed between the AMES and the educational level and age of the participants. 

Finally, an ANOVA was calculated between the AMES and the participants’ contact degree with Maghrebis (*F* = 45.20; *p* < 0.001), from which three groups emerged: on the one hand, those with lower levels of AMES indicated “Almost never” (*M* = 2.64) and “Rarely” (*M* = 2.84), while a second group was made up of those who indicated “Occasionally” (*M* = 3.59) and, lastly, a third group, with the highest levels of AMES, indicated “Frequently” (*M* = 4.08) and “Very frequently” (*M* = 4.18).

## 4. Discussion and Conclusions

The aim of the present study was to create and validate the AMES in the Spanish context. Firstly, descriptive statistics were adequate for each of the seven items developed. It should be mentioned that in order to obtain these results, three items of the scale had to be replaced, considering what also happened in previous research where some items were problematic and had to be removed [47,48]. It is probable that these items had to be replaced with others that were more culturally adjusted to represent the attitudes towards Maghrebis in Spain, and to fit better with the data [49]. Likewise, according to the scale’s validity, the factor structure of the AMES was satisfactorily examined using exploratory factor analysis (EFA) and confirmatory factor analysis (CFA). In this sense, the CFA results showed an adequate fit index for the unidimensional model to the data. In addition, in the present study, the AMES shows an adequate level of reliability for the whole scale as well as for each item. In general, the psychometric properties of the scale yielded values consistent with those presented in other similar scales, but evaluated the diversity and interculturality attitudes toward the migrant population in general [50], instead of assessing one in particular (Maghrebi migrants). Due to all of these reasons, the AMES was found to be a considered feasible measure to apply in the Spanish context. 

Secondly, there were significant positive correlations observed between the AMES and psychosocial variables (EC, contact, warmth and competence) and were negative with SDO, as was expected. On one hand, the positive correlation with EC, contact, warmth and competence, might be helping to regulate emotions and positive attitudes towards the Maghrebi migrants in education [51,52]. In this sense, frequent intergroup contact may increase knowledge and reduce prejudices towards different culturally diverse groups. In the same way, empathic concern, warmth and competence promote a high orientation towards contact with a more prosocial and intercultural character. Therefore, it is possible to foster more favorable attitudes towards the inclusion of excluded groups, and in this case, Maghrebi migrant teachers and students. On the other hand, this may show that when evaluating the specific attitudes towards Maghrebi migrants, the SDO also scores higher in group dominance and in opposition to equality towards this group, in the educational context. These relations are consistent with similar previous studies supporting SDO as a negative psychosocial variable, which is related to attitudes towards sociocultural diversity [53], intergroup contact and, specifically, with the African population [54]. Therefore, social dominance may exercise a mechanism for sustaining the exclusion of Maghrebi teachers and students from the educational system, assuming there is an obstacle to the development of favorable attitudes towards them.

Thirdly, in relation to gender, the current study revealed statistically significant differences in the overall levels of the AMES. This shows that, unlike men, women in the present sample had more favorable attitudes towards the intercultural inclusion of Maghrebi students and teachers in education. This inequality between men and women, demonstrated on the AMES, is compatible with other previous evidence that evaluated similar constructs, which maintains that women had a greater orientation towards interculturality [55]. Interestingly, other sociodemographic variables (age and educational level) are not shown as determinants in the study, thus, we did not find significant differences. This lack of differences contrasts with the findings evidenced in macro-studies carried out in the European context [56], in which it was shown that citizens with higher education levels were more likely to have pro-immigration attitudes and that older respondents also showed fewer attitudes in favor of immigration. The results show this homogeneity in the scores, regardless of the age and educational levels, and could be related to the influence of other psychosocial variables, such as the level of intergroup contact. Thus, it is concluded that the higher is the frequency of contact with the Maghrebi people, the more favorable are the attitudes towards the inclusion of the Maghrebi students and teachers. The present results are in partial accordance with previous research [57] and could provide increasing evidence of the bonds between the AMES assessment and the relation with other psychosocial variables. A variable that has not been considered in our study is the ethnicity of the participants. This aspect would be important for future research, in order to observe if there is a difference in the values of the AMES with respect to the ethnic origin of the participants.

Regarding the implications of the present study, its results contribute to the research on intercultural attitudes towards Maghrebi migrants in education as well as on the other psychosocial variables involved. At present, paying attention to cultural diversity is regarded as a major challenge to strengthen equity and equality [58]. The positive promotion of cultural diversity is therefore considered a guarantee of social cohesion and solidarity, as a response to the need to improve educational and social coexistence. This kind of attitude may also contribute to migration management by the European institutions, since it depends on the positive attitudes toward the reception and integration of migrants within an inclusive educational and citizen approach [59]. Previously, some studies [60,61] had only focused on evaluating teachers regarding their training attitudes, evaluating their positive perceptions regarding the inclusion of interculturality in schools as a pedagogical proposal. However, until today, there are hardly any studies interested in studying the attitudes towards interculturality in other socio-educational agents, such as families and citizenship in general, which also have a notable impact on this process.

Therefore, it is necessary to consider the different psychosocial variables and implications derived from them in the study of attitudes towards migrant diversity in educational communities. The change in the educational paradigm must go through a social reflection; this implies changes inside and outside the classroom about the perspective that education must be assumed as the main tool for social change and for a more intercultural coexistence. The studies that have investigated the attitudes toward migrants show that there are different degrees of threat, depending on their origin and racialization, being more intense with the rejection of Muslims, especially those who are from Maghreb countries [62]. Thus, the contribution of the evaluation instruments offering the possibility to empirically study (inside and outside the educational context) the attitudes towards interculturality towards specific groups such as Maghrebis, is essential. 

## Figures and Tables

**Table 1 ijerph-19-07303-t001:** Items of Attitudes toward Maghrebis in Education Scale (AMES).

	*M*	*SD*	*S*	*K*	*rjx*	*α.-x*
1. Universities should guarantee access to Maghrebi immigrant students.	3.67	1.43	−0.658	−0.883	0.679	0.865
2. The government must guarantee that sons and daughters of Maghrebi immigrants receive the necessary education.	4.29	1.20	−1.652	1.584	0.595	0.875
3. In schools where there are Maghrebi children, the study of the reality of those countries should be encouraged.	3.29	1.50	−0.339	−1.321	0.701	0.862
4. The State should increase the number of Maghrebi immigrant teachers in public schools.	2.65	1.41	0.234	−1.178	0.797	0.850
5. The educational system must promote civic values oriented to the recognition of the Maghrebi culture.	3.05	1.55	−0.113	−1.467	0.784	0.851
6. The majority of Maghrebi students occupy places in public education, worsening the quality of this.	2.27	1.47	0.675	−1.044	0.725	0.859
7. Maghrebi students must be integrated into the school dynamics, since their presence is an accidental fact.	3.56	1.46	−0.590	−1.050	0.434	0.895

Note: *M*: mean; *SD*: standard deviation; *S*: skewness; *K*: kurtosis; *rjx*: item-total correlation; *α.-x*: alpha if item deleted.

**Table 2 ijerph-19-07303-t002:** Fit indices for the attitudes towards Maghrebis in education Scale (AMES).

	χ^2^(df) ***	S-B χ^2^(df) ***	ΔS-B χ^2^(df)	NNFI	CFI	SRMR
AMES	105.418(14)	89.99(14)	6.42	0.936	0.958	0.045

Note. χ^2^(df): chi-square (degrees of freedom); S-B χ^2^(df): Satorra–Bentler chi-square (degrees of freedom); ΔS-B χ^2^(df): Division Satorra–Bentler chi-square degrees of freedom; NNFI: Non-Normed Fit Index; CFI: Comparative Fit Index; SRMR: Standardized Root Mean Square Residual; Adequate fit: S-B χ^2^(df) ≤ 7; NNFI, CFI and IFI ≥ 0.90; SRMR ≤ 0.08. *** *p* < 0.001.

**Table 3 ijerph-19-07303-t003:** Pearson’s correlations between AMES and other variables.

	1	2	3	4	5	6
1. AMES	0.88	−0.655 **	0.440 **	0.361 **	0.744 **	0.657 **
2. SDO		0.89	−0.504 **	−0.291 **	−0.551 **	−0.535 **
3. EC			0.78	0.296 **	0.429 **	0.415 **
4. Contact				0.82	0.401 **	0.393 **
5. Warmth					0.81	0.749 **
6. Competence						0.77

Note: Cronbach’s α for each scale on the diagonal. AMES: Attitudes toward Maghrebis in Education Scale; SDO: social dominance orientation; EC: empathic concern. ** *p* < 0.01.

## Data Availability

The data that support the findings of this study are available from the corresponding author upon reasonable request.

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
