# Peer review of "Construct Validity of the Attitudes towards Maghrebis in Education Scale (AMES)"

_ijerph, 2022, doi:10.3390/ijerph19127303_

Round 1

Reviewer 1 Report

Good day, I will now review your manuscript. Following are my comments:

1.- I find it very well written, I almost usually recommend language check, so it might be worthwhile doing a final review for grammar and spelling, but I can't find evident mistakes. 

2.- Page 2, line 77. "The scale was developed by [14]". I would add the name of the author, even if it's a bit off format. 

3.- Again, line 92.

4.- I would add a paragraph underlining the difference between measuring explicit and implicit attitudes. So as to state out the obvious.

5.- Were participants Spaniards in origin? Any information on ethiticity of said participants?

6.- Instrtuments are detailed in section 1.2, and then again in the Method. Why not concentrate this information so as to save some space?

7.- I'm finding it hard to align the studies objectives with the instruments. Why so many? Obviously, because of the complexity of attitudes. However, I feel this is not detailed in the introduction. 

8.- Table 1 states "self-perception", whilst the items themselves speak of a global perspective in policy and government. Maybe it's the wrong title. The instrument description states otherwise. 

9.- I would state which authors state what adequate fit is (analyses are correct). 

10.- Linear regressions? Structural modeling? Authors are missing the wow point of their study. The data is there, but the study deserves a more thorough presentation of results. 

11.- I wouldn't state the study as a "validation" because it does not cover Cohen and Swerdlik's 5 phase validation process. I would change the title to "influences" or only "construct validation", which it does provide. 

The study is very worthwhile, but I don't feel the manuscript is there yet. I would adjust the aims to a correlation study, more than a validation (which is a better fit). This should aid in the structure of the information presented. 

Reviewer 2 Report

This is a commendable research tool development, and subsequent data analysis, which will have many uses in the study of intercultural attitudes. The authors have concisely presented the process by which they developed and tested the validity of the instrument. The use of a number of statistical methods to validate the test results promotes credibility in the findings and the conclusions which are plausible and interpretated from the findings. Your reasoning for the use of these particular methods is of value to the reader, as are the interpretations of the data. There is sufficient data presented in the tabulated data to enable the audience to understand the paper.

Reviewer 3 Report

The framework made in the introduction is focused on the Spanish reality and with Spanish authors. Perhaps, initially, the state of the art could be framed at an international level.

The point 2.3. could be further developed, namely by clarifying the context and conditions in which the study took place.

P. 4, line 4 (153) replace "has" to "have"

P. 6, 237, replace "y Competent" to "and Competence"

P. 8, 324 and 325 , replace one of the "towards" for a synonym.

Need to review references:

. 1 and 2 - homogenize the way of referencing the author's name

. After ":" the words must appear in capital letters (homogenize)

. In the books, the edition number, publisher, city and country must appear.

. References 14 and 23 refer to which document? a Journal, an Abstract Book, Proceedings, an oral communication? Pages are missing.

. 19 and 22 titles should be in lowercase.

. 23 and 38 (eg.) put in italics the name of the newspaper

Round 2

Reviewer 1 Report

I still feel a final language check is required, I still found grammatical mistakes (i.e. "resident's").